# Stream Flow Generation for Simulating Yearly Bed Change at an Ungauged Stream in Monsoon Region

**Woong Hee Lee [1]** , **Heung Sik Choi [2], Dongwoo Lee [1] and Byungwoong Choi [3],***

1   Smart Platform Team on AST Co., Ltd., Hwaseong 18453, Korea; wh.lee@astkorea.net (W.H.L.); dklee2@astkorea.net (D.L.)
2   Department of Civil Engineering, Sangji University, Wonju 26339, Korea; hsikchoi@sangji.ac.kr
3   Research Team on Ecological and Natural Map, National Institute of Ecology, Seocheon 33657, Korea
*   Correspondence: bchoi@nie.re.kr

**Abstract:** The stream flow generation method is necessary for predicting yearly bed change at an ungauged stream in Monsoon region where there is no hydrologic and hydraulic information. This study developed the stream flow generation method of daily mean flow for each month over a year for bed change simulation at an ungauged stream. The hydraulic geometries of cross-sections and the corresponding bankfull indicators of the Byeongseong river of 4 km reach were analyzed to estimate the bankfull discharge. The estimated bankfull discharge of the target reach was 77.50 m$^3$/s, and the total annual discharge estimated 3720 m$^3$/s through the correlation equation with the bankfull discharge. The measured total annual discharge of the Byeongseong river was 3887.30 m$^3$/s, which is greater by 167.30 m$^3$/s of 4.3% relative error. The volume and bed changes over a year by the Center for Computational Hydroscience and Engineering Two-Dimension (CCHE2D) model simulated using the measured discharge during 2013 and 2014 coincided with the surveyed in the same period. Estimated total annual discharge was used for the scenarios of stream flow generation. The generated stream flow using the flow apportioned to each month on the basis of the flow percentage in an adjacent stream simulated the river bed most appropriately. The generated stream flow using the flow based on the monthly rainfall percentage of the rainfall station in the target stream basin also simulated river bed well, which is confirmed as an alternative. Quantitatively, the root mean square error (RMSE), mean bias error (MBE), and mean absolute percentage error (MAPE) in-depth change of thalweg between the measured and the simulated were found to be 0.25 m, 0.04 m, and 0.44%, respectively. The result of the simulated cross-sectional river bed change for target reach coincided well with the surveyed. The proposed method is highly applicable to generate the stream flow for analyzing the yearly bed change at an ungauged stream in Monsoon region.

**Keywords:** stream flow generation; bankfull discharge; total annual discharge; ungauged stream; Monsoon region

## 1. Introduction

Bed change analyses are classified into short-term bed change analysis, in which the aspects of erosion and deposition including local scours in the surroundings of a stream structure or at a curved channel are analyzed, and long-term bed change analysis, in which the occurrence of erosion and deposition caused by the change in the sediment transport form over time is analyzed. Accurately observed hydraulic data and hydrologic data are important and fundamental to predicting and analyzing bed changes. It is time-consuming and expensive to acquire the hydraulic data and hydrologic data and investigate the bed changes of many streams in various regions. The hydraulic data and hydrologic data are collected mostly in large-scale rivers, and most of small-sized and medium-sized streams are not gauged. Therefore, it is very important to looking for an alternative to replace the adequate hydraulic and hydrologic data instead of the measured for analyzing the corresponding bed changes at ungauged small- and medium-sized streams.

A two- or three-dimensional model is commonly used to analyze the characteristics of local erosion and deposition around river structures, otherwise a one-dimensional model is used for the prediction of long-term bed change. However, long-term bed change analyses using a two- or three-dimensional model have disadvantages in that the simulation takes a long time, and an incorrect simulation result may be obtained unless an accurate boundary condition and sediment transport formulas are used when applying the model. Therefore, most of the bed change analyses using a two- or three-dimensional model are short-term bed-change analyses when flooding.

In Monsoon region, the summer and the fall have a flooding due to the highly concentrated precipitation. The spring and the winter are considered to be dry seasons because there is little precipitation. As a result, seasonal changes over a year evidently affect the river bed. Accordingly, the streams in Monsoon-affected regions show different annual bed changes from the diverse changes in flow. Therefore, it is important to analyze the hydrologic, hydraulic, and bed change during a year in Monsoon region [1,2].

The stream flow dominant to bed change in Monsoon region exhibits substantial seasonal variation of flood and low flows over a year. Thus, it is imperative to predict the long-term bed change rather than the short-term bed change with respect to the flood season for planning various river projects. Accordingly, for the analysis of the long-term bed change, accurate data of the flow and sediment that dominate the bed change should be used. Choi et al. [3] performed a simulation of bed change for 12 years from 1986 to 1998 using the surface-water modeling system (SMS), a two-dimensional bed change model, and the result coincided with the surveyed cross-section, confirming the applicability of the model. Ahn et al. [4] analyzed the erosion and deposition sections of sediment load in the Mississippi river by simulating bed change for approximately 20 years from 1975 to 1995 and comparing the result with the surveyed, which resulted in good agreement. Garcia-Martinez et al. [5] developed an algorithm reducing the simulation time of two-dimensional long-term bed change and verified its applicability by comparing the result of bed-change simulation for the Apure river for 5 years with the result of a one-dimensional model. Klar et al. [6] verified the accuracy of the sediment transport analysis for 10 years by using the distributed computing approach to reasonably reduce the limit on the calculation time of sediment transport simulation using a two-dimensional model. Karmaker and Dutta [7] predicted morphological and thalweg changes in a dynamic braided-river reach, and confirmed the applicability of a 2D model in a complex riverine using numerical riverine model of MIKE21C.

To estimate the stream flow for simulating the bed change in an ungauged stream, the drainage-area ratio method [8] and the regional regression method [9] have been the most popular methods. Several previous studies have estimated the stream flow in an ungauged stream [10–20]. The indirect estimation of stream flow is thought to be more suitable for flood-discharge estimation than drought flow [21–23], and together with the verification of the simulation result, it was emphasized that the accurate and continuous long-term precipitation data should be used. Therefore, the discharge estimated indirectly lacks reliability when used as the discharge condition for a simulation of long-term bed change. Although many studies have been conducted on the estimation of stream flow in an ungagged stream, studies on the estimation of stream flow for the prediction of long-term bed change are scarce.

Bed changes in rivers are governed by the hydraulic geometry, which is known to have a high correlation with the hydraulic characteristics as it has a relation with the hydrologic characteristics of basins [24]. Recently, bankfull discharge was used as the channel-forming discharge that plays a dominant role in bed change and channel forming in a stream. Bankfull discharge, which forms the hydraulic geometry of a stream, is defined as the discharge that transports the maximum sediment load annually [25–31]. The hydraulic geometric indicators of a stream formed by bankfull discharge can be used as important elements for the analysis of the hydrologic and hydraulic characteristics of the stream. The discharge with moderate magnitude and frequency plays a major role in the

similar movement due to the correlation between the dominant flow rate and the similar amount [32], and the flow rate that moves most of the annual similar amount over the years is defined as the effective discharge [32,33]. According to the previous studies, the stable loads that have reached dynamic equilibrium have similar values of bankfull discharge, effective discharge, and frequency discharge with specified recurrence interval [33–35]. Therefore, the bankfull discharge appearing as the river channel formation flow rate can be analyzed by calculating the flow rate transferring the maximum sediment load of the year [33–36].

This study developed the generation method of daily mean flow of each month over a year for simulating the yearly bed change at ungauged stream in Monsoon region. The bankfull discharge of channel forming discharge was estimated by investigating bankfull indicators of the hydraulic geometry proposed by McCandless [37] and then determined the total annual discharge by the relation with bankfull discharge [38]. To find the stream flow for predicting the yearly bed change, various stream flow scenarios of daily, monthly, and seasonal mean discharges including flow duration distribution were generated based on the total annual discharge. In addition, to evaluate the applicability of the Center for Computational Hydroscience and Engineering Two-Dimension (CCHE2D) model, the bed change simulation results using the measured discharge were compared with the surveyed data. The bed change simulation results of the generated stream flow for each scenario were compared with the surveyed. From this, we suggested a new stream flow generation method that can simulate the yearly bed change appropriately in Monsoon region.

## 2. Materials and Methods

### 2.1. Study Area

In Figure 1a, the study area is the Byeongseong river, which is a tributary of the Nakdong river, the 2nd largest river in Korea (N 128°15′74″ E 36°40′8″). The basin area of the Byeongseong river is 434.0 km$^2$, the length of the stream is 30.0 km, and the basin shape factor is 0.42, showing a dendritic channel type.

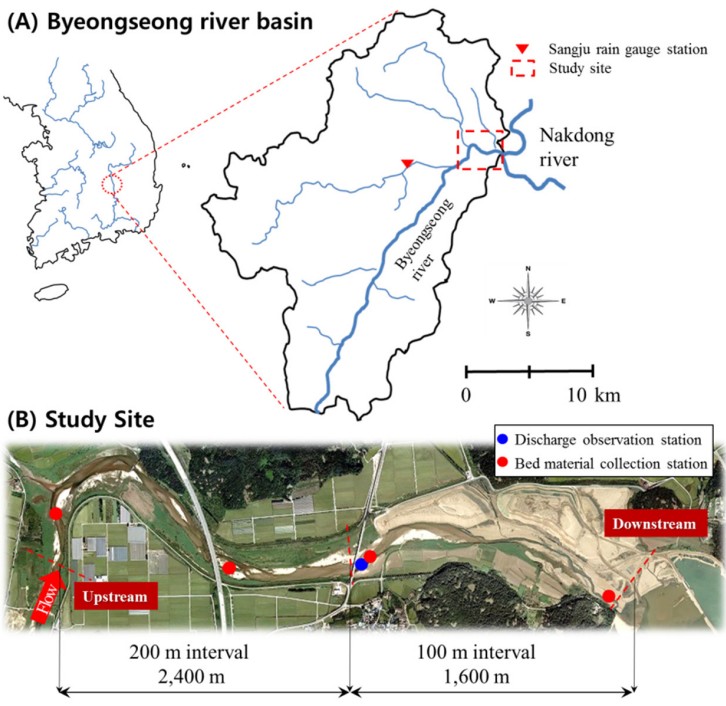

**Figure 1.** Study area: (**A**) Byeongseong river basin, (**B**) study site of the Byeongseong river and cross-section intervals.

In Figure 1b, when selecting the target area for the numerical simulation, it is important to analyze the diverse hydraulic geometry conditions to determine the changes in the flow conditions and to simulate the bed change. In the present study, the hydraulic and bed change characteristics at the Section 4 km upstream from the section flowing into the Nakdong river were analyzed. The average slope of the channel is 0.0015 (m/m) and the average particle size of the bottom substrate is 0.79 mm, which is characteristic of a sand river. In the 4 km of the Byeongseong river target reach has various flow conditions such as bending, ripple, pool, and flow pattern change, which makes it suitable to study and analyze the bed change patterns with respect to the generated stream flow.

The Byeongseong river basin has a rainfall observatory of Sangju Rainfall Station. The mean annual precipitation of the Sangju Rainfall Station basin in 2013–2014 is 1213.0 mm, which is approximately 97.43% of 1245.0 mm, the mean annual precipitation in Korea. The precipitation characteristics of Sangju Rainfall Station from 1981 to 2013 are shown in Figure 1. The precipitation in the Byeongseong river basin clearly shows the characteristics of Monsoon climate, in which 45% or more of the total precipitation is concentrated on July and August.

Figure 2 shows the flow chart outlining the research processes. The research was performed by estimating a bankfull discharge by analyzing the bankfull indicators in the target stream. The total annual discharge was estimated by the relation with the bankfull discharge. We generated annual time series flow scenarios for each month and compared the measured discharge. Then, the corresponding bed change simulation results for each scenario were compared with the surveyed bed change data. By comparing the simulating results of bed change for each scenario with the measured, we proposed the stream flow generation for simulating yearly bed change at an ungauged stream in Monsoon region (Figure 3).

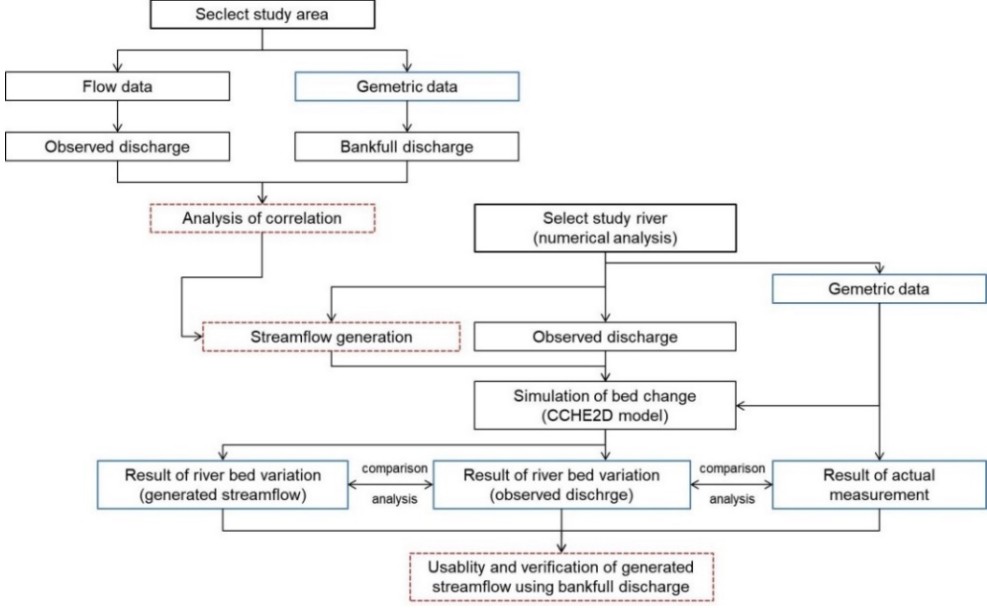

**Figure 2.** Flow chart of stream flow generation for simulating yearly bed change.

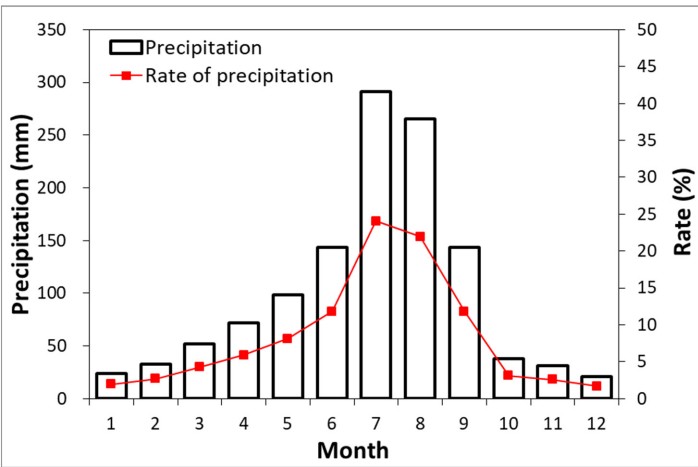

**Figure 3.** Monthly precipitation and rate of precipitation in Sangju Rainfall Station (1981–2013) (Gyeongsangbuk-Do, 2011).

*2.2. Bankfull Discharge Estimation and Surveyed Bed Data*

To analyze the geometric characteristics and the yearly bed change of the section, the 28 cross sectional geometry were surveyed for 1 year from 2013 to 2014. In addition, due to the characteristics of this study the applicability of the CCHE2D model was validated through the comparison between the measured and the simulated results for bed change. The cross sections were surveyed at intervals of 100 m from the channel junction to the 2 km upstream and then at intervals of 200 m until 4 km. Figure 2 shows the surveyed cross-section of the Byeongseong river.

Figure 4 shows the geomorphic bankfull indicators proposed by McCandless [37], which were applied for identifying bankfull cross-section in the present study. The five main geomorphic bankfull indicators are as follows: floodplain break, inflection point, scour line, depositional bench (active channel), and point bar. For estimating the bankfull discharge, one method is based on the stage-discharge curve by confirming the water stage corresponding to the bankfull indicator and another method conducts a numerical simulation using the hydraulic-geometric data in accordance with the water stage corresponding to the bankfull indicator [39]. Accordingly, the discharge at each section that flows up to the water stage corresponding bankfull indicator was presented as bankfull discharge by the Hydrologic Engineering Center's River Analysis System (HEC-RAS) model, one-dimensional numerical model, using the water stage corresponding to the bankfull indicator and the cross-section data (Table 1). The bankfull discharges for each section of the Byeongseong river were estimated to vary between a minimum of 20 m$^3$/s and a maximum of 150 m$^3$/s, and the averaged bankfull discharge of the target reach was 77.50 m$^3$/s.

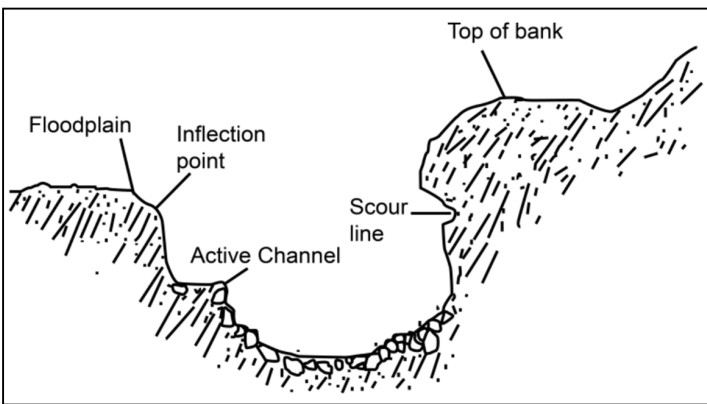

**Figure 4.** Cross-sectional diagram of bankfull indicators.

**Table 1.** Estimated bankfull discharge at the Byeongseong river (bankfull mean depth ($D_{bf}$; m); bankfull width ($W_{bf}$; m); bankfull discharge ($Q_{bf}$; m³/s); bankfull indicator (B.I.)).

| Distance | $D_{bf}$ | $W_{bf}$ | $Q_{bf}$ | B.I. | Distance | $D_{bf}$ | $W_{bf}$ | $Q_{bf}$ | B.I. |
|---------|------|--------|--------|------|---------|------|--------|--------|------|
| 100 | 0.93 | 94.36 | 80.00 | scour line | 1500 | 0.47 | 99.11 | 50.00 | scour line |
| 200 | 0.67 | 108.50 | 80.00 | inflection point | 1600 | 0.90 | 146.53 | 60.00 | point bar |
| 300 | 0.53 | 80.77 | 100.00 | inflection point | 1800 | 1.36 | 63.06 | 60.00 | scour line |
| 400 | 0.91 | 128.02 | 80.00 | inflection point | 2000 | 0.51 | 67.26 | 20.00 | scour line |
| 500 | 1.03 | 142.22 | 80.00 | inflection point | 2200 | 0.57 | 93.36 | 50.00 | point bar |
| 600 | 0.88 | 167.96 | 70.00 | inflection point | 2400 | 0.60 | 112.09 | 125.00 | point bar |
| 700 | 0.63 | 137.96 | 50.00 | point bar | 2600 | 0.86 | 153.55 | 80.00 | point bar |
| 800 | 1.05 | 106.77 | 80.00 | inflection point | 2800 | 1.08 | 29.97 | 110.00 | floodplain break |
| 900 | 1.48 | 89.75 | 110.00 | point bar | 3000 | 1.51 | 106.60 | 150.00 | inflection point |
| 1000 | 0.82 | 86.33 | 90.00 | point bar | 3200 | 1.06 | 88.82 | 110.00 | inflection point |
| 1100 | 0.93 | 72.57 | 70.00 | depositional bench | 3400 | 0.90 | 94.68 | 100.00 | depositional bench |
| 1200 | 0.97 | 65.35 | 110.00 | depositional bench | 3600 | 0.59 | 82.25 | 35.00 | scour line |
| 1300 | 0.32 | 68.64 | 40.00 | depositional bench | 3800 | 0.41 | 89.46 | 20.00 | point bar |
| 1400 | 0.89 | 106.40 | 100.00 | inflection point | 4000 | 0.46 | 62.10 | 60.00 | inflection point |

*2.3. Stream Flow Generation for Bed Change Simulation*

Lee and Choi [38] presented the relation between bankfull discharge and total annual discharge by analyzing the bankfull discharges and measured discharges at 27 points in 19 streams of the Han river basin in Korea, where Monsoon climate prevails. Bankfull discharge is a channel forming discharge that can be defined as having significant effects on the formation of a stream channel. The total annual discharges were calculated by summing the daily mean discharges over a year. The measured daily mean and estimated bankfull discharges indicated that the bankfull discharge was 7.8 times greater than the daily mean discharge, and that the two were highly correlated. This new definition of bankfull discharge for the tributaries of the Han river basin can be employed to reliably determine the bankfull discharge at an ungauged stream using the daily mean or total annual discharge, which can be easily obtained from stream investigations.

Consequently, the total annual discharge estimation method using the relation with bankfull discharge proposed by Lee and Choi [38] was used. The correlation between bankfull discharge and total annual discharge is as follow in Equation (1):

$$Q_{bf} = 7.8 \cdot Q_{day} = Q_m/4 = Q_y/48 \tag{1}$$

where $Q_{bf}$ (m³/s) is the bankfull discharge, $Q_{day}$ (m³/s) is the daily mean discharge, $Q_m$ (m³/s) is the total monthly discharge, and $Q_y$ (m³/s) is the total annual discharge.

The total annual discharge of the target reach of the Byeongseong river was estimated on the basis of Equation (1) using the estimated bankfull discharge. The bankfull discharge of the target reach is 77.50 m³/s, and the total annual discharge estimated accordingly is 3720 m³/s. Furthermore, the measured total annual discharge of the Byeongseong river was 3887.30 m³/s, which is greater by 167.30 m³/s with 4.3% of relative error compared to the total annual discharge estimated using the bankfull discharge.

Six scenarios of daily mean flow discharge for each month were created using the total annual discharge estimated (Table 2). Case I is the daily mean flow discharge over

a month obtained by taking an average of the measured daily mean discharge for each month. Case II is the daily mean discharge for each month obtained by dividing the total annual discharge into 365 days (1 year). The scenarios of cases III to VI were created to enable the flow to be estimated using the hydrologic data and hydraulic data gauged in the surroundings of the target reach, respectively. Precipitation is the most influential factor for river flow, and the rate of monthly effective rainfall is approximately equal to the monthly flow rate of rivers. Therefore, case III is the daily mean flow for each month obtained by dividing the total annual discharge in proportion to the precipitation in each month gauged at the Sangju Rainfall Station located in the Byeongseong river basin. The US Geological Survey reported that the continuously observed data of an existing gauged basin can be expanded as the discharge data of a hydraulically equivalent or similar ungauged stream of the basin [39]. Accordingly, case IV is the daily mean flow for each month obtained by dividing the total annual discharge in proportion to the flow rate in each month measured in the Nakdong river, the main stream of the Byeongseong river. Case V is the daily mean flow for each month obtained by dividing the total annual discharge into each month by using the percentages of flow in the four seasons. In addition, case VI is an apportionment based on flow regime and is the daily mean flow for each month apportioned in the form of a normal distribution based on the percentages of averaged-wet flow ($Q_{95}$), normal flow ($Q_{185}$), low flow ($Q_{275}$), and drought flow ($Q_{355}$).

**Table 2.** Six scenarios of daily mean flow discharge for each month.

| Classification | Explanation | | |
|---|---|---|---|
| Case I | Mean daily discharge by taking an average of the gauged mean daily discharge for each month | | |
| Case II | | Mean daily discharge dividing total annual discharge by 365 days | |
| Case III | Generated mean daily discharge for each month | Monthly precipitation rate at Sangju Rainfall Station | |
| Case IV | | Monthly flow rate at nearby observed stream | |
| Case V | | Seasonal discharge rate in Korea | |
| Case VI | | Flow duration of averaged-wet flow ($Q_{95}$), normal flow ($Q_{185}$), low flow ($Q_{275}$), drought flow ($Q_{355}$) rate | |

Figure 5a,b shows the daily mean flow discharge for each month generated and the measured daily mean flow discharge during 2013 to 2014 at the Byeongseong river. The measured flow was obtained using the stage-discharge rating curve provided by the Water Resources Management Information System (WAMIS) in Korea. In the present study, the methods by using the hydrological regime (magnitude and duration) and seasonally and monthly adjusted minimum flow allocation concepts are used for constructing scenarios. As you can see in Figure 5a, in the Monsoon region, it is very important to analyze seasonal river flow and precipitation variations to prevent the damages of flood in the wet season and the drought in the dry season. Hence, Korea has planning the Water Vision 2020 [40] to analyze the seasonal river flow rate and to manage national water resources.

In Figure 5b show the flow-duration curve is a cumulative frequency curve that show the percent of time specified discharges were equaled or exceeded during period. The flow duration curve is used to determine the optimal design of the river facility and the river management plan. It is used as fundamental data for analyzing the discharge variations of the target stream over a year. The discharge hydrograph of the Byeongseong river was modified to focus on the flood discharge in the summer season (flood season) so that it can match the characteristics of the Monsoon climate.

Table 3 shows the daily mean flow discharge of each month for each scenario and shows the RMSE (root mean square error) with measured flow discharge. The RMSE for case I was 7.89 m$^3$/s and the generated stream flow of the present study was the most similar to the measured flow. The RMSEs for case IV and case III were 9.06 m$^3$/s

and 9.52 m$^3$/s, respectively. The RMSEs for case II, case VI, and case V were 14.80 m$^3$/s, 13.16 m$^3$/s, and 10.65 m$^3$/s, respectively.

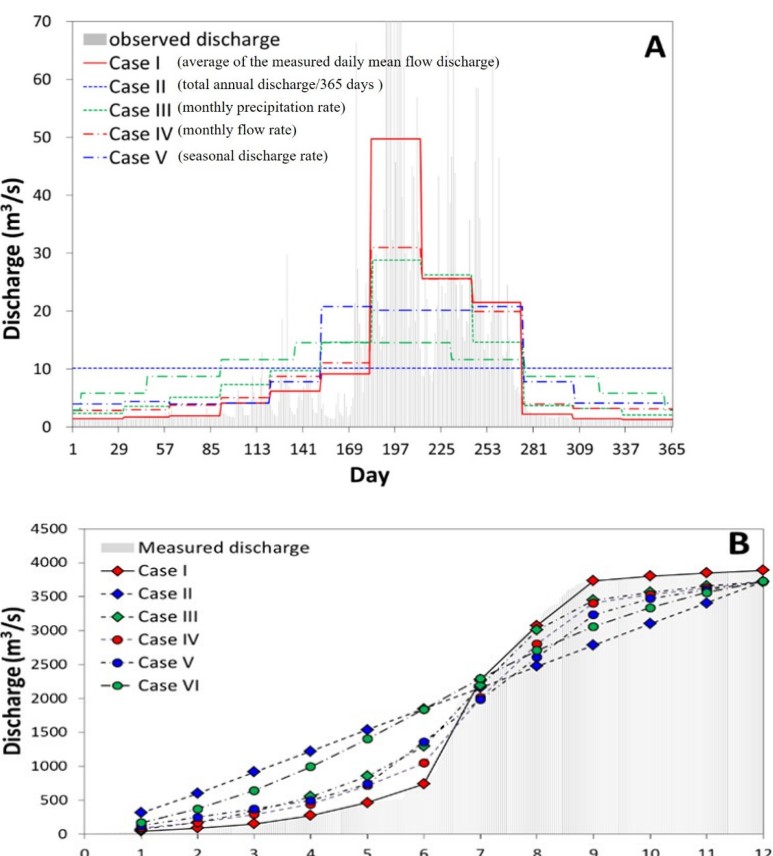

**Figure 5.** Stream flow generated for each case and the measured during 2013 to 2014 at Byeongseong river: (**A**) daily mean discharge (mean discharge from the daily values), (**B**) discharge mass curve (cumulative discharge volume).

**Table 3.** Generated mean daily discharge for each month with six scenarios.

| Month | Mean Daily Discharge for Each Month (RMSE) (m$^3$/s) | | | | | |
|---|---|---|---|---|---|---|
| | **Case I** | **Case II** | **Case III** | **Case IV** | **Case V** | **Case VI** |
| Jan | 1.43 (0.09) | 10.19 (8.90) | 2.36 (0.95) | 2.87 (1.46) | 4.00 (2.61) | 5.37 (4.00) |
| Feb | 1.68 (0.32) | 10.19 (8.67) | 3.56 (1.94) | 3.01 (1.39) | 4.42 (2.81) | 7.30 (5.73) |
| Mar | 1.94 (0.39) | 10.19 (8.40) | 5.13 (3.27) | 3.79 (1.92) | 4.00 (2.13) | 8.76 (6.94) |
| Apr | 4.10 (2.57) | 10.19 (6.71) | 7.33 (4.17) | 5.06 (2.75) | 4.13 (2.57) | 11.68 (8.13) |
| May | 6.16 (5.55) | 10.19 (6.90) | 9.77 (6.65) | 8.76 (6.15) | 7.80 (5.79) | 13.19 (9.05) |
| Jun | 9.13 (13.13) | 10.19 (13.17) | 14.67 (14.29) | 11.13 (13.29) | 20.77 (17.68) | 14.60 (14.26) |
| Jul | 49.68 (36.66) | 10.19 (54.36) | 28.88 (42.32) | 31.07 (41.25) | 20.10 (47.41) | 14.60 (51.14) |
| Aug | 25.57 (18.12) | 10.19 (23.93) | 26.32 (18.13) | 25.59 (18.12) | 20.10 (18.95) | 13.38 (21.95) |
| Sep | 22.05 (16.55) | 10.19 (20.48) | 14.68 (18.17) | 19.95 (16.69) | 20.77 (16.60) | 11.68 (19.63) |
| Oct | 2.23 (0.97) | 10.19 (8.15) | 3.72 (1.80) | 4.01 (2.06) | 7.80 (5.75) | 8.95 (6.90) |
| Nov | 1.46 (0.15) | 10.19 (8.88) | 3.17 (1.74) | 3.24 (1.81) | 4.13 (2.72) | 7.40 (6.04) |
| Dec | 1.31 (0.22) | 10.19 (9.03) | 2.04 (0.77) | 3.10 (1.83) | 4.00 (2.74) | 5.37 (4.13) |
| RMSE * | 7.89 | 14.80 | 9.52 | 9.06 | 10.65 | 13.16 |

* Averaged root mean square error (RMSE) (m$^3$/s).

This is because that case I is a daily mean flow over a month obtained by taking an averaged of the measured daily mean discharge for each month. It is believed that the flow characteristics similar to the actual flow regime. However, it is difficult to acquire actual measured data in ungauged watershed. Douglas et al. [41] performed the applicability of flow data in ungauged region using seasonal drainage-area ratio method in Red River, which are the characteristics of inland climate in the mid-North of the US. However, the results have pointed out that the characteristics of Monsoon region are somewhat different. In this regard, the case IV is in good agreement with measured data with considering the characteristics of the Monsoon climate.

## 3. Results and Discussion

### 3.1. Calibration and Verification of Model

A two-dimensional model (CCHE2D) was used to analyze the hydraulic and bed change characteristics of the target reach. We performed the input parameter calibration as shown in Table 4 and model applicability by comparing the surveyed bed data with the simulated bed change results using the measured discharge of the target reach from 2013 to 2014. We set the time step to 60 s and total simulation time to 31,536,000 s to simulate the yearly bed change. The transport mode was set to the total load as bed load plus suspended load. As shown in Table 4, the input data of representative diameter, roughness coefficient, bed material, and the sediment rating curve (Figure 6) were used reported by Gyeongsangbuk-Do [42]. The sediment rating curve that fit the relationship between the river flow discharge (Q) and the suspended sediment concentration (C) are commonly used to evaluate the patterns and trends of river water quality. In many previous studies, it is assumed that the sediment rating curve has a power-law form. The sediment rating curve is related to the effective discharge. The effective discharge is defined as the flow rate that moves most of the annual similar amount over the years and is calculated using the flow rate-frequency distribution curve and the similar amount curve. The effective discharge is an essential element that can be a standard flow rate in designing the river channel or evaluating the stability of the existing river channel, but it is difficult to derive a specific pattern because the analysis of the current flow rate is not performed sufficiently in Korea. In particular, the rivers in Korea have difficulty in applying foreign cases to Korea because the coefficient of river regime is 10 to 25 times larger than the US and Europe due to the climate characteristic that about 70% of the annual precipitation is concentrated in summer.

**Table 4.** Inputs of CCHE2D model in simulation.

| Parameter | Value |
| --- | --- |
| Turbulence model | Mixing length model |
| Viscosity coefficient | 1 |
| Wall slipness coefficient | 0.50 |
| Threshold value of depth to consider (wet/dry) | 0.04 (m) |
| Roughness coefficient | 0.03 |
| Representative diameter | 0.00079 (m) |
| Define size class | 10 categories in 0.000125 (m)~0.064 (m) |
| Sediment specific gravity | 2.66 |
| Porosity | 0.40 |
| Transport mode | Total load as bed load plus suspended load |
| Diffusivity (Schmidt number) | 0.5 |
| Total simulation time | 31,536,000 (s) |
| Time step | 60 (s) |

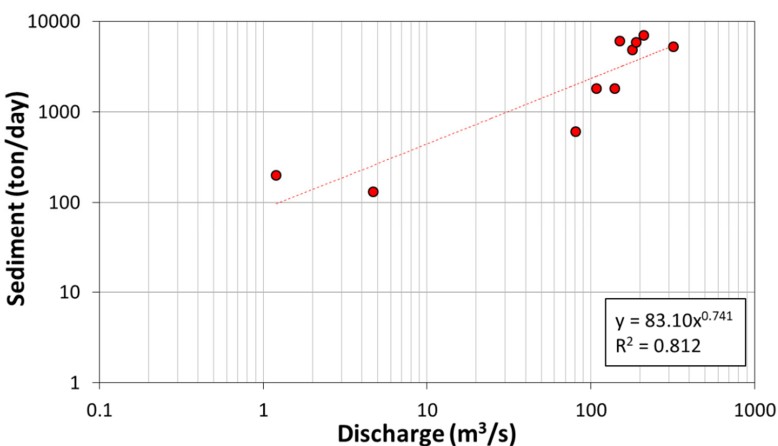

**Figure 6.** Sediment rating curve at study site of the Byeonseong river (Gyeongsangbuk-Do, 2011).

The bed change of the 4 km target reach of the Byeongseong river surveyed in 2013 and 2014 was simulated with the measured inputs of stream flow and channel geomorphic terrain, as shown in Figure 7. The mesh comprised 13,365 nodes at a transverse interval of 20 m and a longitudinal interval of 10 m. For the simulation of the river bed change in the study area, the monitoring data were used twice in 2013 and 2014. The sedimentation pattern generally occurred at 4 km of the target section due to the change of the river bed for one year from 2013 to 2014, and sedimentation occurred at the end of the maximum curve and at the local part at the downstream, and erosion occurred inside the water culvert at 2 km, which was confirmed to have changed into a stable river bed without irregularities overall.

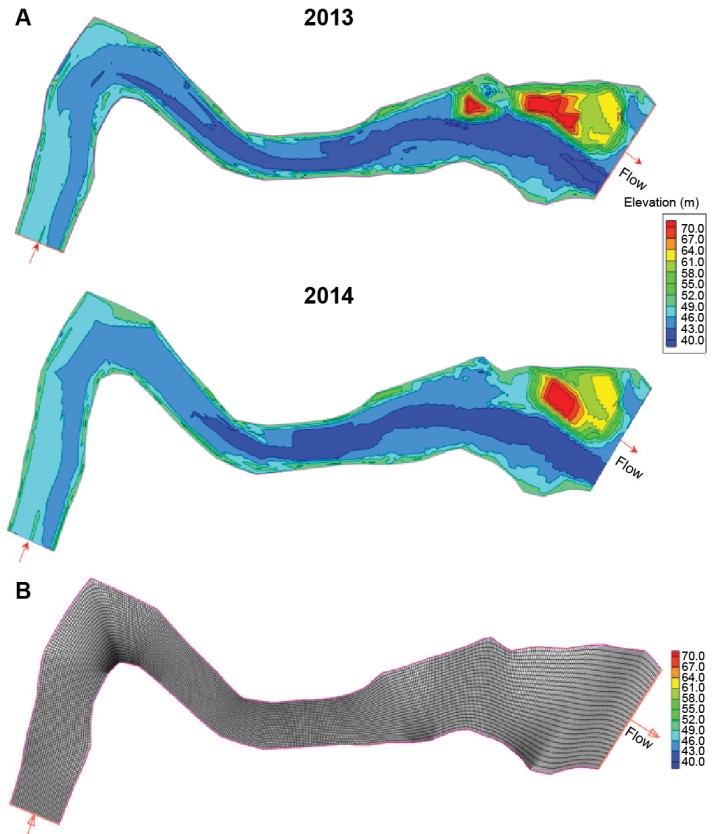

**Figure 7.** Topographic and elevation map of the study area: (**A**) distributions of bed elevation, (**B**) mesh of the study area.

The numerical simulation result of the bed change using the surveyed bed data and the measured discharge in 2013 and 2014 is shown in Figure 8. The simulation results of bed change along thalweg using the measured discharge were similar with the surveyed in 2014. The RMSE, MBE (mean bias error), and MAPE (mean absolute percentage error) were found to be 0.18 m, 0.05 m and 0.36%, respectively. The result of the bed change simulation after calibrating the CCHE2D model parameters coincided well with the characteristics of the bed change in the surveyed bed. It can be seen that the riverbed elevation has deposited in 2014 compared to the riverbed change in 2013. The sediment diameter of the particle was small, so the bed change (erosion and deposition) occurred actively, but it is believed that the riverbed stabilization has progressed since 2014.

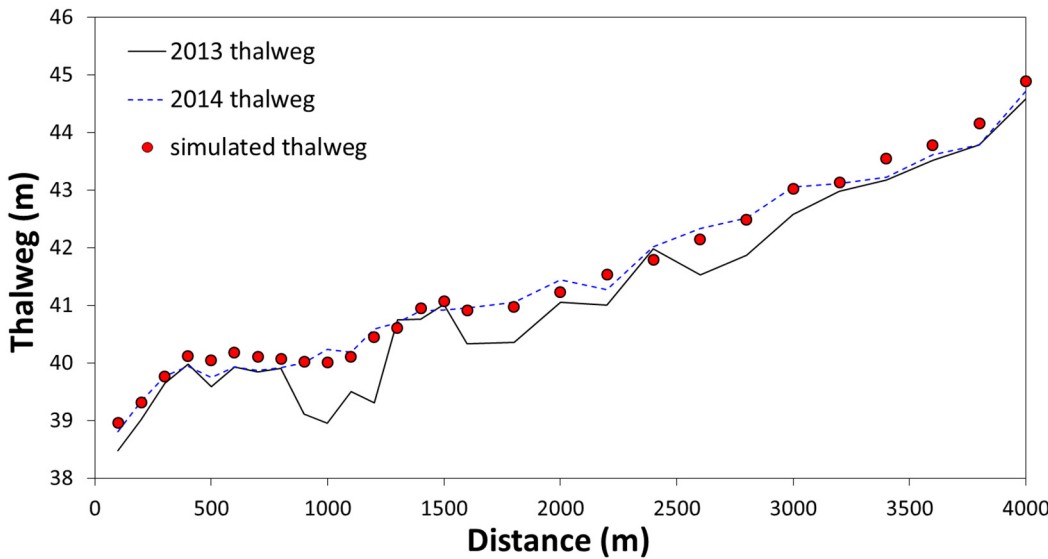

**Figure 8.** Bed change result in 2013 and 2014 according to the measured and the surveyed along thalweg.

### 3.2. Yearly Bed Change Using the Stream Flow for Each Scenario

Figure 9 shows the result of bed change for 1 year for the generated stream flow scenarios. It was found that, in general, the result of analyzing the surveyed along thalweg coincided well with the result of bed change for the six cases of simulation discharge scenarios including the measured stream flow. The result of analyzing the bed change along thalweg showed that significant erosion and deposition occurred between 800 m and 1300 m from the downstream section. The bed change aspects in this section using simulation stream flows showed a large difference from those of the surveyed bed. This difference is thought to be due to the effect of the artificial river bed disturbance caused by bed protection work in the section between 800 m and 1300 m from the downstream section.

Table 5 shows the RMSE, MBE, and MAPE of the bed changes at the 28 cross sections along thalweg between the surveyed and the simulation results of the six simulation stream flow scenarios. The stream flow scenario that most appropriately simulated the characteristics of the bed change was case IV, the RMSE, MBE, and MAPE are 0.21 m, −0.01 m, and 0.40%, respectively. Thus, the apportioning method of generating the stream flow using the percentage of flow at the gauging station in the nearby surrounding river is most appropriate. The RMSE, MBE, and MAPE of the bed change along thalweg between the surveyed and the measured for case III were 0.25 m and 0.04 m and 0.44%, respectively, which were similar to those of case IV. Therefore, the apportioning method of generating the stream flow using the monthly precipitation percentage of the rainfall station in the target stream basin also simulated river bed well, which was confirmed applicable as an alternative. Furthermore, the flow in a flood season is found to be close to the bankfull

discharge in cases III and IV that governs the bed formation because a relatively large flow occurred in this season.

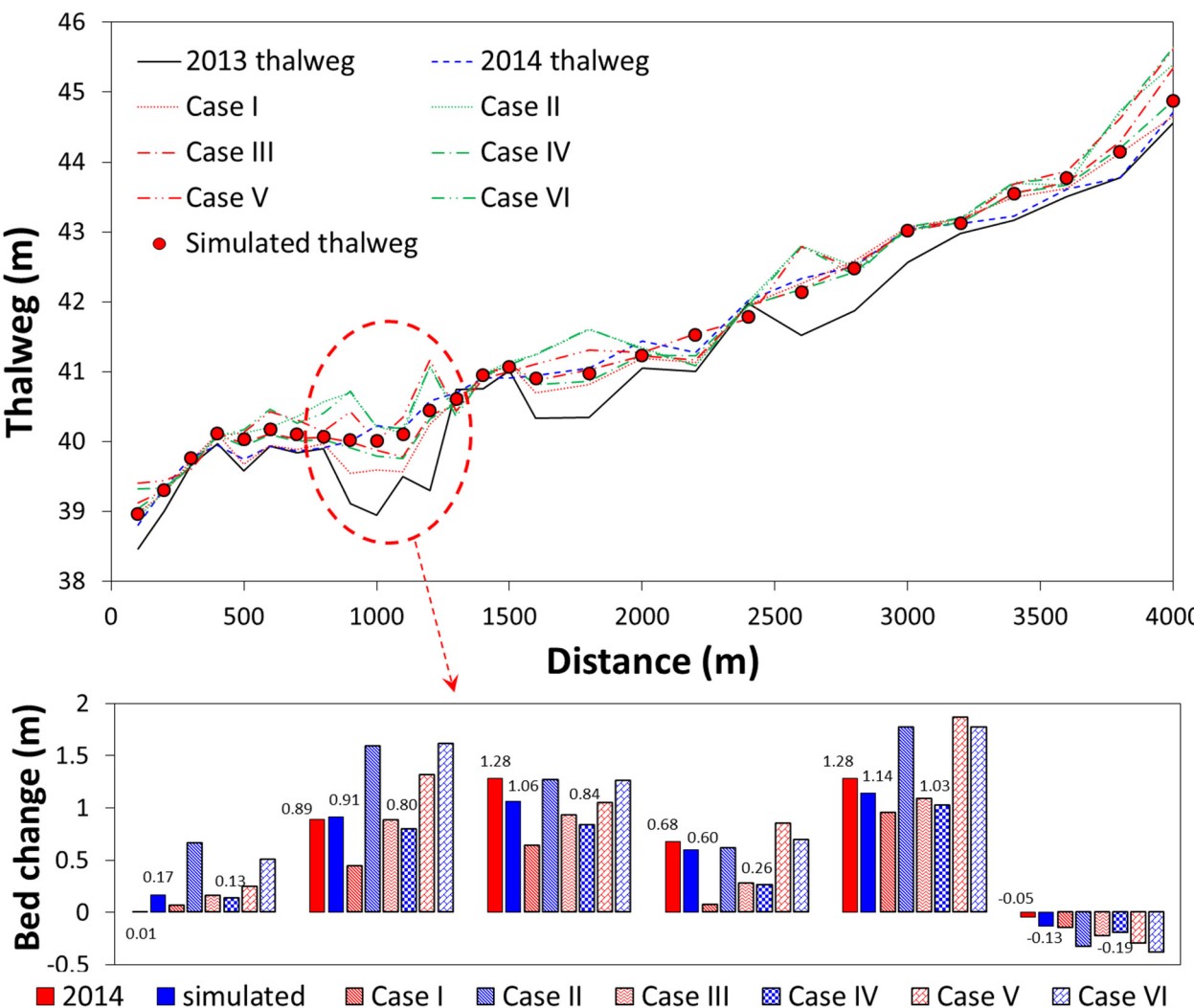

**Figure 9.** Simulated and surveyed results of river bed change along thalweg with six cases of simulation discharge scenarios.

**Table 5.** Errors of river bed change between the simulated and the surveyed at the 28 cross sections.

| Classification | Errors of Bed Change between the Surveyed and the Simulated along Thalweg (2014) | | |
|---|---|---|---|
| | RMSE (m) | MBE (m) | MAPE (%) |
| Case I | 0.25 | −0.07 | 0.42 |
| Case II | 0.40 | 0.23 | 0.69 |
| Case III | 0.25 | 0.04 | 0.44 |
| Case IV | 0.21 | −0.01 | 0.40 |
| Case V | 0.39 | 0.22 | 0.73 |
| Case VI | 0.43 | 0.25 | 0.77 |

Although the magnitudes of monthly stream flow generated do not reach the bankfull discharge of 77.50 m³/s, the simulation results obtained using the generated stream flows coincided with the surveyed results. Consequently, stream flow below bankfull discharge

causes erosion and deposition in a stream over time, leading to a dynamic equilibrium of the stable channel. Therefore, it is necessary to find the governing flow for yearly bed change in an ungauged stream. In cases II, V, and VI, the RMSE, MBE, and MAPE between the simulated and the surveyed were found to be not smaller than 0.39 m, 0.22 m, and 0.69%, respectively. These results were thought to be due to the failure of the generated stream flows to contribute to the formation of the bed which reflect the discharge pattern of the Monsoon climate.

Table 6 shows the volume of bed change and the errors between the measured and the simulated for each scenario. Similar result was shown such that the volumes of bed change simulated by the measured stream flow was 85,904.10 m$^3$ and that by scenario IV was 90,338.96 m$^3$ over a year as shown in the analyses of bed change along thalweg and the RMSE, MBE, and MAPE between them 1026.36 m$^3$, 164.25 m$^3$, and 0.11%, respectively. The volume of bed change simulated by scenario III was 94,878.50 m$^3$ which were similar to those of case IV and the RMSE, MBE, and MAPE errors between the measured and the simulated were 1058.41 m$^3$, 332.38 m$^3$, and 0.12%, respectively. The volume of bed change of scenarios of case II, V, VI were over 17,8926.22 m$^3$ which was shown significant deposition along river reach and the RMSE, MBE and MAPE errors between the measured and the simulated were found to be not smaller than 4675.17 m$^3$, 3445.26 m$^3$, and 0.49%, respectively. The volume of bed change of scenarios of case I was 41,672.60 m$^3$ and the RMSE, MBE, and MAPE errors between the measured and the simulated were 2039.90 m$^3$, $-1638.21$ m$^3$, and 0.27%, respectively. It was shown that the simulation result of scenario IV had good agreement with the surveyed. Thus, the scenario IV of the apportioning method of generating the stream flow using the percentage of flow at the gauging station in the nearby surrounding river is most appropriate.

**Table 6.** Total volume of river bed change and errors between the measured and the simulated both using the surveyed and six scenarios.

| Classification | Errors for Volume of Bed Changes between the Measured and the Simulated Both Using the Measured and the Generated Stream Flows | | | | |
| --- | --- | --- | --- | --- | --- |
| | Total Volume (m$^3$) | Bed Change Volume (m$^3$) | RMSE (m$^3$) | MBE (m$^3$) | MAPE (%) |
| Initial bed (2013) | 16,824,619 | - | - | - | - |
| Measured stream flow | 16,910,523 | 85,904.10 | - | - | - |
| Case I | 16,866,291 | 41,672.60 | 2039.90 | $-1638.21$ | 0.27 |
| Case II | 17,008,044 | 183,425.29 | 4675.17 | 3611.89 | 0.55 |
| Case III | 16,919,497 | 94,878.50 | 1058.41 | 332.38 | 0.12 |
| Case IV | 16,914,958 | 90,338.96 | 1026.36 | 164.25 | 0.11 |
| Case V | 17,003,545 | 178,926.22 | 4770.42 | 3445.26 | 0.49 |
| Case VI | 17,019,878 | 195,259.38 | 5446.68 | 4050.19 | 0.62 |

* No change happens to the current default setting which is now set as a '-'.

Figure 10 shows the bed change aspect of case IV, which most appropriately simulated the change along thalweg on the representative cross sections, was comparatively analyzed with the surveyed cross-section data. The representative cross sections were selected at 4 points. A-A' was an upstream point, B-B' was the point of highest curvature, C-C' was a midstream point, and D-D' was a downstream point. The result of the bed-change simulation using the case IV scenario well reproduced the sediment aspects of the study reach. The RMSEs and MBEs between the measured and the simulated of the cross-section A-A', B-B', C-C', and D-D' were 0.086 m and 0.017 m, 0.041 m and 0.007 m, 0.120 m and 0.040 m, and 0.037 m and $-0.003$ m, respectively. The prediction of yearly bed change using the daily mean flow for each month by apportioning the total annual discharge estimated

using the total monthly discharge percentage of an adjacent stream well simulated the longitudinal and transverse bed changes.

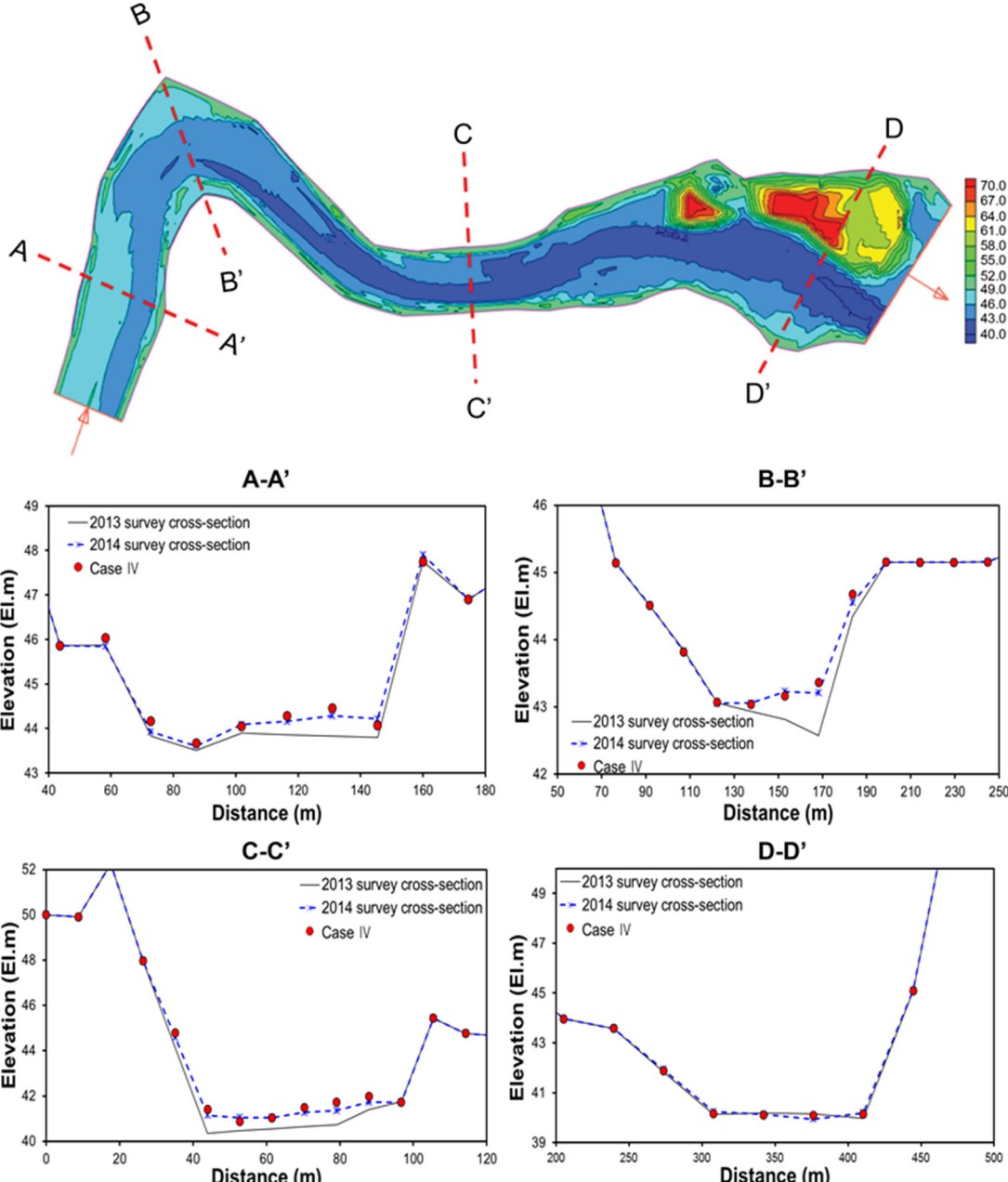

**Figure 10.** Bed change result at representative cross sections aspect of scenario case IV.

## 4. Conclusions

This study developed the generation method of daily mean stream for each month for bed change simulation at an ungauged stream. The total annual discharge was estimated with the relation of the bankfull discharge. Using the estimated total annual discharge, we proposed six stream flow scenarios for simulating yearly river-bed change and proposed the appropriate stream flow generation method to predict yearly river-bed change at an ungauged stream in Monsoon region. The main conclusions of this study are as follows.

- The hydraulic geometries of cross-sections and the corresponding bankfull indicators of the Byeongseong river of 4 km reach were analyzed to estimate the bankfull discharge. The estimated bankfull discharge of the target reach was 77.50 m³/s. The

total annual discharge estimated 3720 m$^3$/s using the correlation equation with the bankfull discharge and the measured total annual discharge of the Byeongseong river was 3887.30 m$^3$/s, which had 4.3% of relative error.

- The bed change was simulated using CCHE2D, a two-dimensional model. The simulation result of bed change using the measured stream flow in a 4 km of the target reach coincided well with the surveyed along thalweg during 2013 and 2014, which showed the applicability of CCHE2D. The RMES, MBE and MAPE between the measured and the simulated bed changes were found to be 0.18 m, 0.05 m, and 0.36%, respectively.

- The generated stream flow of scenario IV using the flow apportioned to each month on the basis of the flow percentage in an adjacent stream simulated the river bed most appropriately. The RMSE, MBE, and MAPE in depth change along thalweg between the surveyed and the simulated were 0.21 m, −0.01 m, and 0.40%, respectively. The simulated total volume of bed change found to be 90,338.96 m$^3$ which was similar with the measured of 85,904.10 m$^3$ and the RMSE, MBE, and MAPE were 1026.36 m$^3$, 164.25 m$^3$, and 0.11%, respectively.

- The generated stream flow of scenario III using the flow based on the monthly rainfall percentage of the rainfall station in the target stream basin also simulated the river bed well. The RMSE, MBE, and MAPE in depth change of thalweg between the measured and the simulated were found to be 0.25 m, 0.04 m, and 0.44% respectively. The result was found to be very similar to scenario IV. Thus, the stream flow generation method of scenario III is an alternative at an ungauged stream where has no information for stream flow.

- The result of the simulated cross-sectional river bed change for a target reach using on the basis of the stream flow generated using the percentage of an adjacent stream coincided well with the surveyed bed.

- It was possible to estimate the bankfull discharge in an ungauged stream by analyzing hydraulic geometry, and the total annual discharge could be estimated using the bankfull discharge. The daily mean flow for each month by apportioning the estimated total annual discharge using the total monthly flow percentage of an adjacent stream were applicable for stream flow generation for simulating the yearly bed change at an ungauged stream in Monsoon region.

**Author Contributions:** Conceptualization, W.H.L., H.S.C., D.L., and B.C.; methodology, W.H.L., H.S.C., D.L., and B.C.; formal analysis, W.H.L., H.S.C., D.L., and B.C.; investigation, W.H.L., H.S.C., D.L., and B.C.; resources, W.H.L., H.S.C., D.L., and B.C.; data curation, W.H.L., H.S.C., D.L., and B.C.; writing—original draft preparation, W.H.L., H.S.C., D.L., and B.C.; writing—review and editing, W.H.L., H.S.C., D.L., and B.C.; visualization, W.H.L., H.S.C., D.L., and B.C.; funding acquisition, W.H.L. and D.L. All authors have read and agreed to the published version of the manuscript.

**Funding:** This work was supported by Korea Environment Industry and Technology Institute (KEITI) through Intelligent Management Program for Urban Water Resources Project, funded by Korea Ministry of Environment (MOE) (2019002950005).

**Institutional Review Board Statement:** Not applicable.

**Informed Consent Statement:** Informed consent was obtained from all subjects involved in the study.

**Data Availability Statement:** The data presented in this study are available on justified request from the first or corresponding authors.

**Acknowledgments:** The authors would like to thank AST Co., Ltd. for its support in this work.

**Conflicts of Interest:** The authors declare no conflict of interest.

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
