# Peer review of "Stream Flow Generation for Simulating Yearly Bed Change at an Ungauged Stream in Monsoon Region"

_water, doi:10.3390/w13040554_

Round 1

Reviewer 1 Report

A high quality graphical abstract is required for better understanding of research process.

Figure 1 is not clear, please use a better picture for this aim.

Introduction section is not enough as a high-quality research. Literature review is very poor. Please edit it and fix it by more literature reviews.

Caption of Figures should be developed, please provide more explanation for each tables and figure in caption.

Do not put figures continuously, each figure needs explanation separately (this is very important).

Figure 5, 6 , 7, 8, 9 must be discussed more, please explain them clearly.

Discussion part should be compared by other studies result.

It is not clear how the author used the Monsoon Region! Please describe more about it and have a clarification.

Author Response

Reply to Reviewer #1

Authors are grateful for Reviewer #1 for his thorough review. The comments are believed to be very helpful in improving the quality of the manuscript. We tried to reflect all issues raised by the reviewer in the revised manuscript. We highlighted the part we modified in the revision. The followings are our replies to each issue:

Specific comments:

  1. A high quality graphical abstract is required for better understanding of research process.

Following the reviewer’s comments, we added a high quality graphical abstract. Also, we included explanations. Thank you for the positive opinion on our work.

  1. Figure 1 is not clear, please use a better picture for this aim.

Following the reviewer’s comment, we now checked and modified figure. Thank you very much.

  1. Introduction section is not enough as a high-quality research. Literature review is very poor. Please edit it and fix it by more literature reviews.

Thank you for the comments. In the revised manuscript, we added a new paragraph on the detailed bankfull discharge, effective discharge, and frequency discharge effects on the dynamic equilibrium. We added the paragraph below in the introduction:

In Monsoon region, the summer and the fall have a flooding due to the highly concentrated precipitation. The spring and the winter are considered to be a dry season because there is little precipitation. As a result, seasonal changes over a year evidently affect the river bed. Accordingly, the streams in Monsoon-affected regions show different annual bed changes from the diverse changes in flow. Therefore, it is important to analyze the hydrologic, hydraulic, and bed change during a year in Monsoon region [1-2].

Bed changes in rivers are governed by the hydraulic geometry, which is known to have a high correlation with the hydraulic characteristics as it has a relation with the hydrologic characteristics of basins [24]. Recently, bankfull discharge was used as the channel-forming discharge that plays a dominant role in bed change and channel forming in a stream. Bankfull discharge, which forms the hydraulic geometry of a stream, is defined as the discharge that transports the maximum sediment annually [25-31]. The hydraulic geometric indicators of a stream formed by bankfull discharge can be used as important elements for the analysis of the hydrologic and hydraulic characteristics of the stream. The discharge with moderate magnitude and frequency plays a major role in the similar movement due to the correlation between the dominant flow rate and the similar amount [32], and the flow rate that moves most of the annual similar amount over the years is defined as the effective discharge [32-33]. According to the previous studies, the stable loads that have reached dynamic equilibrium have similar values ​​of bankfull discharge, effective discharge, and frequency discharge with specified recurrence interval [33-35]. Therefore, the bankfull discharge appearing as the river channel formation flow rate can be analyzed by calculating the flow rate transferring the maximum amount of the year [33-36].

  1. Caption of Figures should be developed, please provide more explanation for each tables and figure in caption.

Thank you very much. Following the reviewer’s comments, we added and changed explanation for each tables and figures in caption.

  1. Do not put figures continuously, each figure needs explanation separately (this is very important).

Following the reviewer’s comments, we now added clear explanation separately in each table and figure. In addition, we highlighted the explanation and discussion. Thank you very much.

  1. Figure 5, 6 , 7, 8, 9 must be discussed more, please explain them clearly.

Following the reviewer’s comments, we now added and highlighted clear explanation in each figure. In addition, we re-organize the explanation and discussion as suggested by the reviewer. Now, readers can better understand.

  1. Discussion part should be compared by other studies result. It is not clear how the author used the Monsoon Region! Please describe more about it and have a clarification.

Following the reviewer’s comments, we now added and highlighted clear explanation in each figure. In addition, we re-organize the explanation and discussion as suggested by the reviewer. Now, readers can better understand.

The followings are added statements:

This is because that Case I is a daily mean flow over a month obtained by taking an averaged of the measured daily mean discharge for each month. It is believed that the flow characteristics similar to the actual flow regime. However, it is difficult to acquire actual measured data in ungauged watershed. Douglas et al. [41] performed the applicability of flow data in ungauged region using seasonal drainage-area ratio method in Red River, which are the characteristics of inland climate in the mid-North of the US. However, the results is pointed out that the characteristics of Monsoon region are somewhat different. In this regard, the case IV is in good agreement with measured data with considering the characteristics of the Monsoon climate.

Reviewer 2 Report

The manuscript focused on stream flow generation method to assess channel change at an ungauged river in Monsoon region. This issue is important because in many areas the system of hydrological and meteorological measurements is insufficient to study the changes in the channels, especially in areas with varying precipitation throughout the year. However, the quality of the manuscript needs to be improved.

I recommend 'major review'.

The following are some specific comments:

  1. In general the introduction needs some restructure so that in the end you can identify your research gaps and come up with clear research questions. The citation should be supplemented. Please use more international literature.
  2. What is the research hypothesis?
  3. Some issues are unclear to the reader, e.g. total annual discharge, it is not the same that annual runoff. Please explain.
  4. In the manuscript there is no discussion. Please show your research in a broader context.
  5. Please check grammar and typos as well as sentence structure etc.

Detailed comments and suggestions have been marked in the manuscript.

Author Response

Reply to Reviewer #2

Authors thank Reviewer #2 for the thorough and insightful review. Comments made by the reviewer were really helpful in improving the manuscript. We tried to reflect all comments in the revision. The modified parts were highlighted in the revised manuscript. The followings are our replies to each issue:

General comments

The manuscript focused on stream flow generation method to assess channel change at an ungauged river in Monsoon region. This issue is important because in many areas the system of hydrological and meteorological measurements is insufficient to study the changes in the channels, especially in areas with varying precipitation throughout the year. However, the quality of the manuscript needs to be improved.

Thank you for the positive opinion on our work.

Specific comments:

  1. In general the introduction needs some restructure so that in the end you can identify your research gaps and come up with clear research questions. The citation should be supplemented. Please use more international literature.

Thank you for the comments. In the revised manuscript, we added a new paragraph on the detailed bankfull discharge, effective discharge, and frequency discharge effects on the dynamic equilibrium. We added the paragraph below in the introduction:

In Monsoon region, the summer and the fall have a flooding due to the highly concentrated precipitation. The spring and the winter are considered to be a dry season because there is little precipitation. As a result, seasonal changes over a year evidently affect the river bed. Accordingly, the streams in Monsoon-affected regions show different annual bed changes from the diverse changes in flow. Therefore, it is important to analyze the hydrologic, hydraulic, and bed change during a year in Monsoon region [1-2].

Bed changes in rivers are governed by the hydraulic geometry, which is known to have a high correlation with the hydraulic characteristics as it has a relation with the hydrologic characteristics of basins [24]. Recently, bankfull discharge was used as the channel-forming discharge that plays a dominant role in bed change and channel forming in a stream. Bankfull discharge, which forms the hydraulic geometry of a stream, is defined as the discharge that transports the maximum sediment annually [25-31]. The hydraulic geometric indicators of a stream formed by bankfull discharge can be used as important elements for the analysis of the hydrologic and hydraulic characteristics of the stream. The discharge with moderate magnitude and frequency plays a major role in the similar movement due to the correlation between the dominant flow rate and the similar amount [32], and the flow rate that moves most of the annual similar amount over the years is defined as the effective discharge [32-33]. According to the previous studies, the stable loads that have reached dynamic equilibrium have similar values ​​of bankfull discharge, effective discharge, and frequency discharge with specified recurrence interval [33-35]. Therefore, the bankfull discharge appearing as the river channel formation flow rate can be analyzed by calculating the flow rate transferring the maximum amount of the year [33-36].

  1. What is the research hypothesis?

The hypothesis that the dimensions and shapes of channels are determined by events occurring at, or near, the bankfull stage is embodied is much recent work on river morphology.

  1. Some issues are unclear to the reader, e.g. total annual discharge, it is not the same that annual runoff. Please explain.

Following the reviewer’s comments, we now added and highlighted clear explanation in each figure. In addition, we re-organize the explanation and discussion as suggested by the reviewer. Now, readers can better understand.

Fig. 5(a) and Fig. 5(b) shows the daily mean flow discharge for each month generated and the measured daily mean flow discharge during 2013 to 2014 at the Byeongseong River. The measured flow was obtained using the stage-discharge rating curve provided by the Water Resources Management Information System (WAMIS) in Korea. As you can see in the Fig. 5(a), in the Monsoon region, it is very important to analyze seasonal river flow and precipitation variations to prevent the damages of flood in the wet season and the drought in the dry season. Hence, Korea has planning the Water Vision 2020 [40] to analyze the seasonal river flow rate and to manage national water resources.

In the Fig. 5(b) show the flow-duration curve is a cumulative frequency curve that show the percent of time specified discharges were equaled or exceeded during period. The flow duration curve is used to determine the optimal design of the river facility and the river management plan. It is used as fundamental data for analyzing the discharge variations of the target stream over a year. The discharge hydrograph of the Byeongseong River was modified to focus on the flood discharge in the summer season (flood season) so that it can match the characteristics of the Monsoon climate.

This is because that Case I is a daily mean flow over a month obtained by taking an averaged of the measured daily mean discharge for each month. It is believed that the flow characteristics similar to the actual flow regime. However, it is difficult to acquire actual measured data in ungauged watershed. Douglas et al. [41] performed the applicability of flow data in ungauged region using seasonal drainage-area ratio method in Red River, which are characteristics of inland climate in the mid-North of the US. However, the results is pointed out that the characteristics of Monsoon Region are somewhat different. In this regard, the case III is in good agreement with measure data with considering the characteristics of the monsoon climate.

The sediment rating curve that fit the relationship between the river flow discharge (Q) and the suspended sediment concentration (C) are commonly used to evaluate the patterns and trends of river water quality. In many previous studies, it is assumed that the sediment rating curve has a power-law form. The sediment rating curve is related to the effective discharge. The effective discharge is defined as the flow rate that moves most of the annual similar amount over the years and is calculated using the flow rate-frequency distribution curve and the similar amount curve. The effective discharge is an essential element that can be a standard flow rate in designing the river channel or evaluating the stability of the existing river channel, but it is difficult to derive a specific pattern because the analysis of the current flow rate is not performed sufficiently in Korea. In particular, the rivers in Korea have difficulty in applying foreign cases to Korea because the coefficient of river regime is 10 to 25 times larger than the US and Europe due to the climate characteristic that about 70% of the annual precipitation is concentrated in summer.

  1. In the manuscript there is no discussion. Please show your research in a broader context.

Following the reviewer’s comments, we now added and highlighted clear explanation in each figure. In addition, we re-organize the explanation and discussion as suggested by the reviewer. Now, readers can better understand.

  1. Please check grammar and typos as well as sentence structure etc.

- [L18] total annual discharge

Following the reviewer’s comment, we added the definition of total annual discharge in the Materials and Methods

- [L33-L37] introduction

Following the reviewer’s comments, we re-organize the explanation and discussion as suggested by the reviewer. Now, readers can better understand.

- [L38&L44] bed

We changed bed to riverbed. Thank you very much.

- [L47-L50] introduction

Following the reviewer’s comments, we re-organize the explanation and discussion as suggested by the reviewer. Now, readers can better understand.

- [L56&L59] references

Following the reviewer’s comments, we added references. Thank you very much.

- [L90-L91] introduction

- [L106] abbreviation

Following the reviewer’s comment, we checked abbreviation. Thank you very much.

- [L133] unit

We added.

- [L154&L158] modified sentence

Following the reviewer’s comments, we re-organize the explanation. Now, readers can better understand.

- [L256] modified sentence

We changed. Thank you very much.

- [L265] reference

We changed.

- [L322] definition

Following the reviewer’s comment, we added the sentence to Materials and Methods.

- [L340] unit

We added. Thank you very much.

- [L495] number

We added.

Reviewer 3 Report

Please check the attached comments.

Author Response

Reply to Reviewer #3

Authors are grateful for Reviewer #3 for his thorough review. The comments are believed to be very helpful in improving the quality of the manuscript. We tried to reflect all issues raised by the reviewer in the revised manuscript. We highlighted the part we modified in the revision. The followings are our replies to each issue:

General comments

The author has developed the stream flow generation method of daily mean flow for each month over a year for bed change simulation at an ungauged stream. The topic is interesting for the journal's readership. Also, the methods are right and enough and the results are very good and reasonable. However, in my opinion some issues should be addressed and improved before the manuscript can be published.

 Thank you for the positive opinion on our work.

Specific comments:

  1. Line 20: please give the full name about “CCHE2D model” at the first time.

Following the reviewer’s comment, we checked abbreviation. Thank you very much.

  1. Line 25: How well? Could you please give the actual value?

Following the reviewer’s comment, we added following statement:

Quantitatively, the RMSE, MBE, and MAPE in depth change of thalweg between the measured and the simulated were found to be 0.25 m, 0.04 m, and 0.44% respectively.

  1. Lines 53-58: Please add some references:

Following the reviewer’s comment, we added. Thank you very much.

Kyoung, Min Soo, Hung Soo Kim, Bellie Sivakumar, Vijay P. Singh, and Kyung Soo Ahn. "Dynamic characteristics of monthly rainfall in the Korean Peninsula under climate change." Stochastic Environmental Research and Risk Assessment 25, no. 4 (2011): 613-625.

Duan, W., He, B., Nover, D., Fan, J., Yang, G., Chen, W., ... & Liu, C. (2016). Floods and associated socioeconomic damages in China over the last century. Natural Hazards, 82(1), 401-413.

  1. Line 106: please give the full name about “CCHE2D model” at the first time.

Following the reviewer’s comment, we checked abbreviation. Thank you very much.

  1. Figure 1 shows the research flow, so please move it to Methods.

Following the reviewer’s comment, we changed and modified. Thank you very much.

  1. Lines 139-140: Please give the time period.

Following the reviewer’s comment, we added. Thank you very much.

  1. Tables: please adjust the table according to the regulation of the WATER.

Following the reviewer’s comment, we changed all tables. Thank you very much.

  1. Line 197: Where >> where

We changed. Thank you very much.

9. It is better to discuss the uncertainty about your results.

We totally agree with your opinion. This present study is limited to suggest a new stream flow generation method that can simulate the yearly bed change appropriately in Monsoon region. We will consider uncertainty analysis in future and reflect them in our research. Thank you for your advice.

Round 2

Reviewer 1 Report

Remove vertical lines from the tables

Figure 5 (A) is not clear. Please re-draw it in a better format.

Indicate the latitude and longitude of the study area on the map or in the text.

Author Response

Reply to Reviewer #1

Authors are grateful for Reviewer #1 for his thorough review. The comments are believed to be very helpful in improving the quality of the manuscript. We tried to reflect all issues raised by the reviewer in the revised manuscript. The followings are our replies to each issue:

Specific comments:

  1. Remove vertical lines from the tables

Following the reviewer’s comments, we removed vertical lines from the all tables. Thank you very much.

  1. Figure 5 (A) is not clear. Please re-draw it in a better format.

Following the reviewer’s comments, we now added and highlighted clear explanation in the figure. Now, readers can better understand.

  1. Indicate the latitude and longitude of the study area on the map or in the text.

Following the reviewer’s comments, we added the latitude and longitude of the study area in the revised manuscript. Thank you very much.

Reviewer 2 Report

I accept changes and additions

Author Response

Reply to Reviewer #2

General comments

I accept changes and additions

Thank you for the positive opinion on our work. Authors are grateful for Reviewer #2 for his/her thorough review. The comments are believed to be very helpful in improving the quality of the manuscript.
